# Considerations for Using Neuroblastoma Cell Lines to Examine the Roles of Iron and Ferroptosis in Neurodegeneration

**DOI:** 10.3390/cells13181541

**Published:** 2024-09-13

**Authors:** Cameron J. Cardona, Yoo Kim, Winyoo Chowanadisai, McKale R. Montgomery

**Affiliations:** Department of Nutritional Sciences, Oklahoma State University, Stillwater, OK 74078, USA; cameron.cardona@okstate.edu (C.J.C.); yoo.kim@okstate.edu (Y.K.); winyoo.chowanadisai@okstate.edu (W.C.)

**Keywords:** iron metabolism, cell differentiation, Alzheimer’s disease, cell death, system x_c_-, iron–sulfur cluster biogenesis

## Abstract

Ferroptosis is an iron-dependent form of programmed cell death that is influenced by biological processes such as iron metabolism and senescence. As brain iron levels increase with aging, ferroptosis is also implicated in the development of age-related pathologic conditions such as Alzheimer’s disease (AD) and related dementias (ADRD). Indeed, inhibitors of ferroptosis have been shown to be protective in models of degenerative brain disorders like AD/ADRD. Given the inaccessibility of the living human brain for metabolic studies, the goal of this work was to characterize an in vitro model for understanding how aging and iron availability influence neuronal iron metabolism and ferroptosis. First, the human (SH-SY5Y) and mouse (Neuro-2a) neuroblastoma lines were terminally differentiated into mature neurons by culturing in all-trans-retinoic acid for at least 72 h. Despite demonstrating all signs of neuronal differentiation and maturation, including increased expression of the iron storage protein ferritin, we discovered that differentiation conferred ferroptosis resistance in both cell lines. Gene expression data indicates differentiated neurons increase their capacity to protect against iron-mediated oxidative damage by augmenting cystine import, and subsequently increasing intracellular cysteine levels, to promote glutathione production and glutathione peroxidase activity (GPX). In support of this hypothesis, we found that culturing differentiated neurons in cysteine-depleted media sensitized them to GPX4 inhibition, and that these effects are mitigated by cystine supplementation. Such findings are important as they provide guidance for the use of in vitro experimental models to investigate the role of ferroptosis in neurodegeneration in pathologies such as ADRD.

## 1. Introduction

Iron accumulation occurs in the brain during normal aging and is exacerbated in age-related pathologic conditions such as Alzheimer’s disease (AD) and related dementias (ADRD) [1,2]. Higher levels of brain iron accumulation are also associated with reduced motor ability, increased rate of cognitive decline, as well as degree of cognitive impairment [3,4]. Thus, strategies for mitigating iron-mediated cellular damage could be used to promote healthy aging and protect against neurodegenerative decline. One such approach may be through the prevention of ferroptosis, a form of iron-mediated programmed cell death. Ferroptosis is driven by extensive iron-dependent accumulation of lipid reactive oxygen species (ROS), which ultimately commits cells to death [5]. As such, iron chelators and antioxidants can protect against ferroptosis, whereas loss of lipid peroxide repair pathways and accumulation of excess redox-active iron can induce it [6]. Yet, the mechanisms contributing to this iron imbalance remain unknown.

Given the inaccessibility of the living human brain to metabolic studies, the use of in vitro models is critical for understanding how metabolic changes and nutrient availability influence neuronal aging and neurodegeneration. In this regard, cultured neuroblastoma cell lines have vast experimental value as they can be terminally differentiated into mature neurons with the same morphologic, neurologic, and metabolic characteristics as those in adult brains [7,8,9]. Importantly, as in an aged brain, differentiated neuroblastoma cell lines also appear to accumulate iron [8], and upregulate pathways involved in the evasion of cell death [10], but the mechanisms controlling iron metabolism and homeostasis in undifferentiated versus differentiated cultured neuroblastoma cells remain to be elucidated. 

Contributing to our lack of understanding is the fact that there is no established protocol for investigating the role of iron metabolism in neurodegeneration in cultured neuroblastoma cell lines. The paucity of data that is available is a mixed collection of experiments conducted in either undifferentiated or differentiated cells, and review articles attempting to summarize the major findings from these works often use these data interchangeably to make broad conclusions. Indeed, within the current literature, there are studies suggesting that neurons are exquisitely sensitive to ferroptosis [11], largely resistant to ferroptosis [12], and perhaps only susceptible to a necrotic type of cell death that only resembles ferroptosis [13]. Thus, in this work, we sought to characterize the differences in iron metabolism and ferroptosis susceptibility in undifferentiated versus differentiated cells in two commonly used neuroblastoma cell lines. Our findings highlight the importance of distinguishing between undifferentiated and differentiated cell types and demonstrate that differentiated neuroblastoma cell lines are highly resistant to ferroptotic cell death. 

Despite its name, more than just an overabundance of intracellular iron is necessary for ferroptosis to occur. Particularly, ferroptosis also requires the presence of oxidizable phospholipids and defective, or inhibited lipid peroxide repair. The system x_c_-/glutathione peroxidase 4 (GPX4)/glutathione axis is particularly critical for preventing the accumulation of toxic lipid ROS and protecting against ferroptosis [14]. At the cell membrane, system x_c_- imports extracellular cystine which then gets rapidly reduced to cysteine and can subsequently be used for glutathione biosynthesis and GPX4 production. As GPX4 is the most dominant antioxidant for detoxifying intracellular lipid ROS, inhibition of any part of this pathway can lead to ferroptosis [6]. Herein, we present novel findings that upregulation of system x_c_- may be one mechanism by which aging neurons might protect themselves against iron-mediated cell damage and death. 

## 2. Materials and Methods

### 2.1. Cell Culture and Viability Assays 

SH-SY5Y and Neuro-2a cells (ATCC; Manassas, VA, USA) were maintained at a constant 5% CO_2_, 37 °C, and 95% humidity in Dulbecco’s Modified Eagle’s Media (DMEM) (Corning; New York, NY, USA) supplemented with 10% fetal bovine serum (Atlanta Biologicals; Norcross, GA, USA), 100 µg/mL streptomycin, and 100 IU/mL penicillin. Both cell lines were differentiated into a mature neuron-like phenotype by incubating for at least 72 h in media containing 20 µM all-trans retinoic acid (ATRA) (Sigma-Aldrich, St. Louis, MO, USA). ATRA-induced differentiation was selected because its known growth-inhibiting and differentiation-promoting properties make it one of the most well-characterized methods for the induction of differentiation of both SH-SY5Y and Neuro-2a cell lines [7,15,16]. For experiments comparing differentiated versus undifferentiated cell responses, cells were plated such that they would reach ~85% confluency following 3 days of culture in either untreated or ATRA-containing media. Due to differences in doubling times between SH-SH5Y and Neuro-2a cells, to achieve ~85% confluency following 3 days in culture, SH-SY5Y cells were initially plated at 50% confluency whereas Neuro-2a cells were plated at 25% confluency. For investigation of response to iron availability, following 3 days in culture with or without ATRA, cells were then treated with either the iron chelator desferrioxamine (DFO), the heme iron supplement hemin, or the ferroptosis inducing drugs erastin or RSL3 (Selleck Chemicals; Houston, TX, USA) at the indicated doses. 

To examine the effects of cysteine depletion on ferroptosis sensitivity in differentiated and undifferentiated cells, SH-SY5Y cells were cultured in DMEM or cysteine and methionine-depleted DMEM (ThermoFisher, Waltham, MA, USA, 210132024) (CMD) with or without 20 µM ATRA for 72 h prior to treatment with the indicated doses of RSL3. However, we found that Neuro-2a cells were exquisitely sensitive to cysteine and methionine restriction and that their viability was reduced to <50% in less than 48 h of culturing in CMD media alone (Appendix A). To overcome this limitation, Neuro-2a cells were cultured in DMEM with or without 20 µM ATRA for 72 h as above, and then half of the cells were switched to ATRA containing CMD media at the same time they were treated with the indicated doses of RSL3. To demonstrate the effects of CMD media were due to cysteine and not methionine restriction, rescue experiments were performed by co-treating the cells with 200 µM L-cystine (Sigma Aldrich, St. Louis, MO, USA), which is consistent with physiologic levels available in vivo [17]. 

Differences in sensitivity to treatment with DFO, hemin, or RSL3 between undifferentiated and differentiated cells were determined by metabolic cytotoxicity assay at 24 and 48 h post-treatment, using a synergy H1 microplate reader (Biotek; Winooski, VT, USA). To address the inherent limitations of metabolic cytotoxicity assay, cells were also imaged at the end of each treatment time point on a Keyence BZ-X700 fluorescence microscope (Keyence Life Sciences; Osaka, Japan) using a 20× objective lens. Given the nature of our treatments (i.e., nutrient restriction or supplementation), the accompanying micrograph images allowed us to visually confirm cell death versus a simple reduction in cell proliferation or metabolic activity. 

### 2.2. RNA Isolation and Real-Time Quantitative PCR (RT-qPCR) 

Total RNA was isolated from SH-SY5Y or Neuro-2a cells following differentiation and/or treatment with DFO, hemin, or RSL3 using TRIzol reagent, per the manufacturer’s instructions. The integrity and purity of RNA samples were verified using agarose gel electrophoresis and a NanoDrop One (ThermoFisher, Waltham, MA, USA), respectively, before DNAse treating 1 µg of RNA per sample with a Roche DNAse I recombinant treatment kit (Millipore Sigma, MA, USA). After DNAse treatment, samples were reverse transcribed into cDNA using a Superscript IV RTase kit (ThermoFisher, Waltham, MA, USA). The resultant cDNA was then used to determine differences in relative mRNA expression using SYBR green chemistry and a Bio-Rad CFX Opus 384 Real Time PCR System (Bio-Rad; Hercules, CA, USA). All RT-qPCR experimental data was analyzed using the 2^−∆∆Ct^ method following normalization to Peptidylprolase Isomerase B (*PPIB*) for SH-SY5Y cells or Ribosomal protein L19 (*Rpl19*) for Neuro-2a cells [4]. Primer sequences used in these studies are listed in Table 1 and Table 2 and were either found in the literature or designed using Primer Express v2.0 software. 

### 2.3. Protein Isolation and Expression Analyses 

Total cell protein was isolated by lysing cell pellets on ice through a series of vigorous vortexes in radioimmunoprecipitation assay (RIPA) buffer (50 mM Tris-HCl, pH 8.0, 1% NP-40, 0.5% Na-deoxycholate, 0.1% SDS, 2 mM EDTA, 2 M NaCl, 1 mM DTT, 0.1 mM PMSF, 1× Halt protease inhibitor cocktail (ThermoFisher, Waltham, MA, USA)). The homogenate was then centrifuged at 14,000× *g* for 20 min at 4 °C. The concentration of the protein-containing supernatant was determined using Pierce’s BCA Assay Kit (ThermoFisher, Waltham, MA, USA) prior to being subjected to examination of protein expression by Western blot. 

A total of 30 µg of protein was diluted into 1× Laemmli sample buffer and loaded into a Bio-Rad (Hercules, CA, USA) pre-cast 4–20% Mini-PROTEAN TGC stain-free gel. Following SDS-PAGE separation, the protein was transferred onto a 0.2 µm PVDF membrane using a wet transfer method. Proper transfer of protein to the membrane was verified using Ponceau S stain prior to blocking for one hour in TBS with 0.1% Tween-20 in five percent non-fat dry milk, overnight incubation in Rabbit anti-ferritin-H (FTH1) or anti-actin beta (ACTB) and one-hour in anti-rabbit HRP linked antibody (all: Cell signaling; Danvers, MA, USA). SuperSignal West PICO Plus HRP-activated chemiluminescent substrate (ThermoFisher, Waltham, MA, USA) was then used to visualize protein expression in a ChemiDoc imaging system (Bio-Rad) or exposed to green X-ray film (Thomas Scientific, Swedesboro, NJ, USA). Band density was quantified using ImageJ version 1.54 [28]. Full, uncropped blot images are available in Appendix A.

### 2.4. Statistics

Differences between group means were determined using Student’s *t*-test and one-way or two-way ANOVA with Tukey’s honest significant difference (HST) post hoc test. Experiments with two groups were analyzed using the *t*-test, and those with three or more groups were analyzed using one- or two-way ANOVAs depending on the number of factors included. All ANOVAs were further analyzed using Tukey’s HSD. For experiments involving differences in the effects of CMD on viability, each media vehicle served as its own control due to differences in baseline viability. Because of this, each group was analyzed using a separate one-way ANOVA. All statistical analysis was conducted using GraphPad Prism version 10.0.0 (GraphPad Software, Boston, MA, USA) or R. R Version 4.3.1 (2023-06-16 ucrt) with the tidyverse, ggplot2, dplyr, and multcompView packages [29,30,31,32].

## 3. Results

### 3.1. Differentiated Neuroblastoma Cell Lines Accumulate Ferritin and Are Responsive to Changes in Iron Chelation

To demonstrate the validity of utilizing ATRA differentiated neuroblastoma cell lines as a model system for interrogating neuronal iron metabolism and susceptibility to ferroptosis, we first tested whether differentiated neuroblastoma cells were sensitive to changes in iron availability. To do so, we treated undifferentiated and differentiated cells with either an iron chelator (deferoxamine; DFO), an iron supplement (hemin), or the potent ferroptosis-inducing drug, RSL3, and measured ferritin heavy chain (ferritin; FTH1) protein expression as a readout for responsiveness iron availability. The differentiation status of both SH-SY5Y and Neuro-2a cells was confirmed visually by the transition from rounded cells that tend to cluster to more pyramidal-shaped cells with extended neurites that do not cluster (Figure 1A–D). As expected, both differentiated and undifferentiated SH-SY5Y and Neuro-2a cells decreased ferritin expression in response to iron chelation (Figure 1E,F). However, the addition of supplemental iron only increased ferritin protein expression in undifferentiated SH-SY5Y cells (Figure 1E) whereas increased ferritin expression in hemin-treated Neuro-2a cells did not reach statistical significance (Figure 1F, *p* = 0.058). This may be because differentiation alone was sufficient to significantly increase ferritin expression in SH-SY5Y cells (Figure 1C,E). Differentiation also led towards a trend in increasing ferritin levels in Neuro-2a cells as well (Figure 1D,F), *p* = 0.06. RSL3 treatment had also led to mixed results, with ferritin levels tending to be unchanged by RSL3 treatment in both sets of undifferentiated cells, whereas RSL3 treatment led to decreased ferritin protein expression in differentiated cells. 

### 3.2. Neuronal Differentiation Does Not Influence Tolerance to Changes in Iron Availability

To determine if differences in molecular responsiveness to iron availability influence susceptibility to iron deficiency or iron toxicity, the viability of differentiated and undifferentiated cells was measured following treatment with increasing doses of DFO or hemin, respectively. The lack of a further increase in ferritin expression in differentiated cells with 50 µM hemin treatment suggested that differentiated cells were already at their maximal iron storage capacity (Figure 1C–F). Thus, we hypothesized that differentiated cells would be more sensitive to iron toxicity. However, treatment with up to 80 µM hemin for 48 h did not have a major effect on either differentiated or undifferentiated cell viability (Figure 2A,B). Similarly, iron chelation did not have a markedly different impact on differentiated versus undifferentiated cell viability in SH-SY5Y cells (Figure 2C), whereas undifferentiated Neuro-2a cells were significantly more sensitive to iron chelation than undifferentiated Neuro-2a cells (Figure 2D). It is worth noting that Neuro-2a cells were more sensitive to DFO treatment than SH-SY5Y cells. These findings are likely attributed to the much more rapid proliferation rates of undifferentiated Neuro-2a cells compared to their differentiated counterparts and SH-SY5Y cells and the importance of iron to support such rapid proliferation. Representative micrograph images of each undifferentiated and differentiated cell type treated with the highest dose of hemin (80 µM) or DFO (100 µM) are shown at the right of each bar graph (Figure 2A–D) to corroborate our conclusion that neuronal differentiation does not largely influence tolerance to changes in iron availability. 

### 3.3. Differentiated Neuroblastoma Cell Lines Are Remarkably Resistant to Ferroptosis

To test our hypothesis that differentiated SH-SY5Y and Neuro-2a cells would be more sensitive to ferroptosis, both differentiated and undifferentiated cells were treated with increasing doses of the ferroptosis-inducing drugs erastin or RSL3, and differences in cell viability were measured at 48 h. These drugs were selected because of their defined mechanisms of action and specificity for ferroptosis induction [5]. Whereas erastin functions by directly blocking system x_c_- and subsequently inhibiting cystine import, RSL3 functions by inhibiting GPX4 activity [33]. Unexpectedly, neither undifferentiated nor differentiated SH-SY5Y cells or Neuro-2a cells were sensitive to erastin treatment (Appendix A). Moreover, both differentiated cell lines were markedly more resistant to RSL3-mediated cell death than their undifferentiated counterparts. Differentiated SH-SY5Y and Neuro-2a cells required more than 10-fold and 2-fold higher doses of RSL3, respectively, to significantly reduce their viability to that of their undifferentiated counterparts (Figure 3A–H). 

Another unanticipated observation was the finding that while undifferentiated cells could be rescued from RSL3-mediated cell death by co-treatment with the potent antioxidant and ferroptosis inhibitor, liproxstatin-1 (Lip-1), differentiated cells could not. As a negative control, cells were also co-treated with the apoptosis inhibitor Z-VAD-FMK (Z-VAD), which failed to rescue either the differentiated or undifferentiated cells from RSL3-mediated cell death. These findings suggest that the undifferentiated cells were indeed succumbing to ferroptotic cell death, but the differentiated cells were neither dying from ferroptosis nor apoptosis. To address the limitations of using a resazurin-based cell viability assay to measure cell death as a readout, we also took images of the RSL3 treated cells at each, which clearly illustrate dead and dying cells in both differentiated and undifferentiated cell types, albeit with contrasting morphologic appearances. Whereas the undifferentiated cells at the highest RSL3 dose examined exhibit signs of cell enlargement (Figure 3C,G), the differentiated cells exhibited signs of cell shrinkage (Figure 3D,H). 

### 3.4. Neuronal Maturation Increases the Expression of mRNA Associated with Ferroptosis Resistance

To elucidate mechanisms that could be contributing to reduced ferroptosis sensitivity in differentiated cells, we examined changes in iron- and ferroptosis-related mRNA expression following ATRA-mediated neuronal differentiation. Changes in iron-related mRNA expression between subtypes were variable, with ferroportin (*SLC40A1*; herein referred to as *FPN*) and nuclear receptor coactivator 4 (*NCOA4*) mRNA expression modestly decreasing in SH-SY5Y cells following differentiation, while ferropotin expression increased and *NCOA4* expression was unchanged in differentiated Neuro-2a cells (Figure 4A,B). Decreased expression of *NCOA4* in SH-SY5Y, but not Neuro-2a cells may explain why ferritin protein expression was significantly higher in differentiated SH-SY5Y cells but not differentiated Neuro-2a cells (Figure 1C–F). Other contrasting observations were that expression of the antioxidant heme oxygenase 1 (*HMOX1*) was increased in differentiated SH-SY5Y cells, but the increase did not reach statistical significance in Neuro-2a cells, whereas transferrin receptor (*TFRC*) mRNA expression was statistically significantly increased only in differentiated Neuro-2a cells (Figure 4A,B). 

Two changes that were consistent upon differentiation of both cell types, however, were increased mRNA expression of the cystine co-importer, solute carrier family 7 member 11 (*SLC7A11*), and the potent antioxidant, glutathione peroxidase 4 (*GPX4*) (Figure 4A,B). Such findings suggest that cystine/cysteine-mediated antioxidant production may be contributing to reduced ferroptosis sensitivity in differentiated neuroblastoma cell lines. However, the functional relevance of increased mRNA expression in cell lines alone is not sufficiently compelling evidence to warrant such a strong conclusion, nor does mRNA expression data alone confer sufficient confidence as to the functional and translational relevance of cystine import or cysteine accumulation. Therefore, we utilized the Metabolome Atlas of the Aging Mouse Brain, developed by the UC Davis West Coast Metabolomics Center, to ascertain how aging impacts brain levels of both cysteine and cystine (Figure 4C and Appendix A, respectively). These data indicate that the accumulation of both cystine and cysteine in the brain is a component of the natural aging process, at least in mice. 

### 3.5. Cysteine Depletion Sensitizes Differentiated Neurons to RSL3-Mediated Cell Death 

To determine if increased cystine import and intracellular cysteine are contributing to ferroptosis resistance in differentiated neurons, both SH-SY5Y cells and Neuro-2a cells were grown in cysteine- and methionine-depleted media (CMD) with and without ATRA treatment. Our first observation was that Neuro-2a cells were particularly sensitive to cysteine restriction, as they could not be cultured for longer than 24 h in CMD media without significant impacts on cell viability (Appendix A). SH-SY5Y cell proliferation was somewhat slowed in CMD media, but major impacts on cell viability were not observed even after 72 h of cysteine restriction (Appendix A). As such, SH-SY5Y cells were grown with (differentiated) or without (undifferentiated) ATRA in regular DMEM or CMD media for 72 h prior to treatment with RSL3. Neuro-2a cells, however, were grown in regular DMEM with (differentiated) or without (undifferentiated) ATRA for 72 h, and then half the cells were transitioned to CMD media at the same time as the RSL3 treatment. Cell viability was assessed at 48 h for SH-SY5Y cells and 24 h for Neuro-2a cells. Cysteine restriction sensitized both differentiated SH-SY5Y and Neuro-2a cells to RSL3 treatment to near that of their corresponding undifferentiated cell types (Figure 5A,B). Importantly, we show that the increased RSL3 sensitivity in differentiated cells is specific to cysteine restriction as supplementation with 200 µM cystine conferred ferroptosis resistance. For these purposes, cystine was supplemented rather than cysteine directly due to its increased stability and solubility, and rapid conversion to cysteine once inside the cell [34,35,36]. The 200 µM dose was selected because it matches the concentration of L-cystine in the standard DMEM the control cells were grown in [37]. As seen in our previous experiments (Figure 3B,E), liproxstatin-1 treatment failed to rescue differentiated cells from RSL3-mediated cell death, suggesting that these cells are at least not exclusively dying from ferroptosis. 

## 4. Discussion

In this study, we characterize the distinct morphologic, molecular, and iron-metabolizing phenotypes of neuroblastoma cell lines in both their undifferentiated and differentiated states. In agreement with previous studies, we show that ATRA-differentiated neuroblastoma cell lines more closely resemble mature neurons and provide a better representation of neuronal iron metabolism that would be observed in an adult human brain [7,8,9]. Thus, neuroblastoma cell lines can serve as a model for exploring the relationship between iron metabolism and neurodegeneration, but they should be used in their differentiated state. 

Using ATRA differentiated neuroblastoma cells as a model, we establish that despite increased iron stores, differentiated neurons are remarkably resistant to ferroptosis, which may be attributed to the upregulation of system x_c_- mediated cystine import. This hypothesis is supported by our findings that cysteine depletion sensitizes differentiated neurons to RSL3-mediated cell death, but cystine supplementation confers resistance. Such protective mechanisms are important due to the importance of iron and ferritin accumulation in brain development and for the support of neuronal activity [8] and are exemplified by the fact that neuroferritinopathy leads to neuronal senescence and ferroptotic cell death [38]. However, the signals that promote increased cystine import and the mechanisms by which cysteine confers ferroptosis resistance remain to be fully described. 

The most obvious mechanism by which ATRA-mediated upregulation of cystine import via system x_c_- would protect against neuronal ferroptosis would be by promoting antioxidant activity via increased glutathione/GPX4 production. Indeed, in a recent publication by Tschuck and colleagues, the antioxidant activity of ATRA was demonstrated as essential for neuronal differentiation [39]. Specifically, ATRA increased GPX4 production and decreased lipid peroxidation, which was critical for cell survival during the neuronal differentiation process. ATRA has also been shown to induce the expression of GPX4 as well as HMOX1 and subsequently suppress ferroptosis in the LPS-induced model of liver damage indicating that antioxidant activities of ATRA are not cell-type specific [40]. Beyond its antioxidant-inducing capacities, however, ATRA-mediated cystine import could also indirectly influence cellular iron metabolism. 

For example, due to cysteine’s role as the primary source of sulfur for iron–sulfur (Fe-S) cluster biogenesis, cysteine depletion can also lead to decreased expression and activity of Fe-S cluster-containing proteins [41,42,43]. The resulting Fe-S cluster deficit can subsequently lead to mitochondrial dysfunction and decrease ferroptosis susceptibility, particularly in cell types that rely heavily on oxidative phosphorylation for energy production [42]. In this regard, the fact that a complete switch to oxidative phosphorylation is a prerequisite for neuron differentiation [10] may also be an important consideration when investigating how iron and ferroptosis contribute to neurodegeneration. Death, as a result of perturbed Fe-S biogenesis and mitochondrial dysfunction, instead of ferroptosis would explain our findings that differentiated neurons grown in CMD were rescued by the addition of extracellular cystine, but not antioxidant supplementation. 

Our work is not the first to suggest that differentiated neuroblastoma cell lines are remarkably resistant to ferroptosis. In accordance with our findings, Ito et al. also reported that SH-SY5Y cells are insensitive to erastin treatment, but susceptible to RSL3-mediated cell death [13]. The authors went on to show that RSL3-induced cell death could be partially inhibited by necrostatin-1 and 3,3′-diindolylmethane, but not Z-VAD, suggesting that RSL3 might actually induce a type of necrotic cell death that is similar, but distinct from ferroptosis, in SH-SY5Y cells. Contrarily, others have reported findings suggesting that differentiated SH-SY5Y cells do display some level of ferroptosis sensitivity [11,12]. However, it is worth noting that the doses of RSL3 and erastin used in those studies were quite high and cell death was minimal. Presumably differentiated neuroblastoma cell lines are more sensitive to RSL3 treatment (i.e., GPX4 inhibition) rather than erastin (the system x_c_- inhibitor) because these cells already have higher basal levels of cysteine due to the upregulation of SLC7A11 during the differentiation process. Future studies should be aimed at interrogating this hypothesis. The mechanisms augmenting system x_c_- upregulation during neuronal differentiation and their capacity to protect against both age- and disease-related iron accumulation are of particular interest and should be the subject of future investigations.

## 5. Conclusions

Evidence suggests neurons acquire additional iron early [8] and continue to accumulate iron throughout the lifespan [2,44]. Thus, we hypothesized ferroptosis could contribute to age-related neurodegeneration. To test this hypothesis, we used an in vitro model of neuronal differentiation to interrogate how iron accumulation during and after the differentiation process influenced neuronal sensitivity to ferroptosis. Instead, under standard growth conditions, differentiated neuroblastoma cell lines are well adapted to handle an abundance of intracellular iron and instead may represent a novel model for studying ferroptosis resistance. Beyond its role in antioxidant production, cysteine is also essential for Fe-S cluster biogenesis, and deficits in Fe-S biogenesis alone are sufficient to influence mitochondrial respiration and susceptibility to ferroptosis [41,42]. In this study, we demonstrate that cysteine availability is critical for the health of differentiated neuroblastoma cell lines, particularly in the face of antioxidant insult. These findings are strengthened by metabolomics studies indicating that brain cysteine levels increase dramatically with advanced age [45]. Thus, life-long corresponding increases in cysteine availability may be a means to enhance aged neurons’ capacity to safely handle elevated levels of iron and confer protection against iron- and ROS-mediated pathologies observed in ADRD. 

## Figures and Tables

**Figure 1 cells-13-01541-f001:**
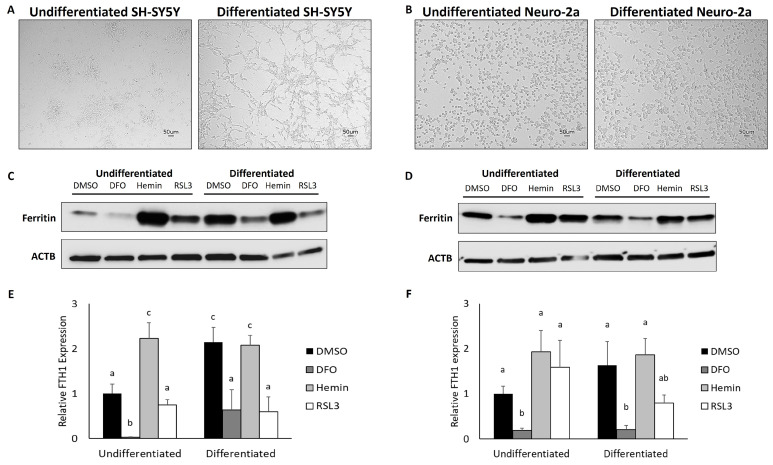
ATRA treatment leads to neuronal differentiation and increases ferritin protein expression in SH-SY5Y and Neuro-2a neuroblastoma cell lines. (**A**) Representative images of SH-SY5Y and (**B**) Neuro-2a cell neurite extension following treatment with ATRA for 72 h. Ferritin expression was assessed in total protein isolated from differentiated and undifferentiated (**C**) SH-SY5Y and (**D**) Neuro-2a cells treated with 50 µM DFO, 50 µM hemin, or 5 µM RSL3 for 24 h and quantified in (**E**,**F**). All microscopy images here and throughout were taken using a BZ-X700 microscope. Differing superscripts (^a–c^) indicate differential expression as analyzed by two-way ANOVA using Tukey’s post hoc test with differentiation status and treatment as factors (*p* < 0.05). Error bars represent ± SEM.

**Figure 2 cells-13-01541-f002:**
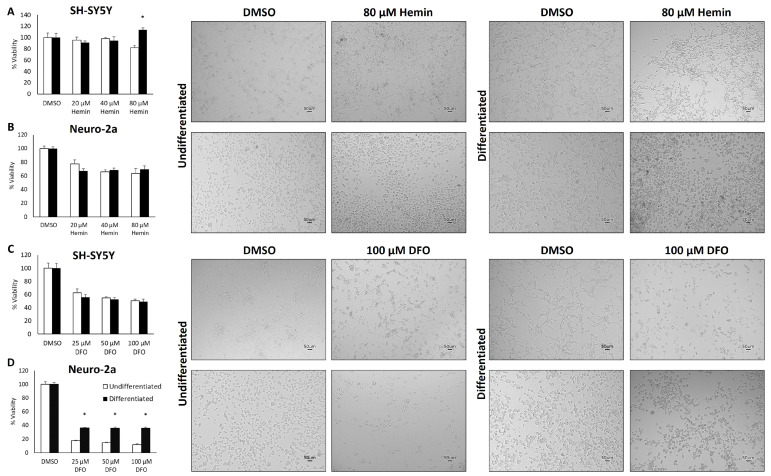
The influence of iron supplementation and iron chelation on cell viability of undifferentiated and differentiated neuroblastoma cell lines. Relative cell viability of (**A**) SH-SY5Y and (**B**) Neuro-2a cells treated with the indicated doses of hemin, and (**C**) SH-SY5Y and (**D**) Neuro2-a cells treated with the indicated doses of DFO for 48 h. Pictures are representative images of undifferentiated and differentiated SH-SY5Y and Neuro-2a cells 48 h after the indicated treatment and just prior to assessment of cell viability. * Indicates differential expression from undifferentiated cells at the same dose as analyzed by Student’s *t*-test (*p* < 0.05).

**Figure 3 cells-13-01541-f003:**
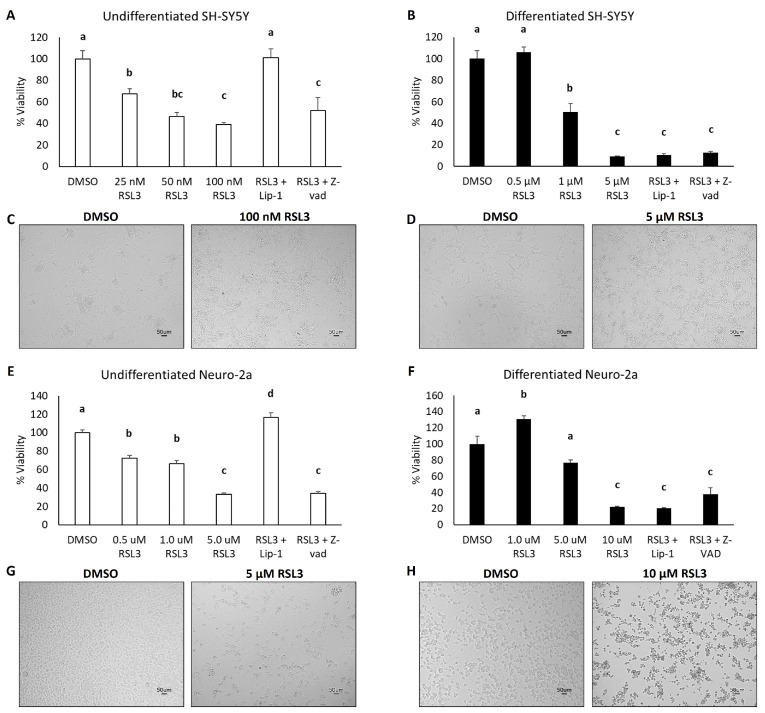
The effect of neuroblastoma cell line differentiation on the susceptibility to RSL3-mediated ferroptotic cell death. Relative cell viability of (**A**,**E**) undifferentiated or (**B**,**F**) differentiated SH-SY5Y and Neuro-2a cells, respectively, in response to treatment with the indicated doses of the potent ferroptosis-inducing drug, RSL3, for 48 h. To indicate that undifferentiated cells were indeed dying from ferroptosis, cells were rescued by treatment with 200 nM of the ferroptosis inhibitor, liproxstatin (Lip-1). Notably, the apoptosis inhibitor, Z-VAD-FMK (Z-VAD) did not rescue either undifferentiated or differentiated cells (**E**,**F**). Representative images of undifferentiated (**C**,**G**) and differentiated (**D**,**H**) SH-SY5Y and Neuro-2a cell lines following 48 h of RSL3 treatment. Differing superscripts indicate differential expression as analyzed by one-way ANOVA using Tukey’s post hoc test (*p* < 0.05).

**Figure 4 cells-13-01541-f004:**
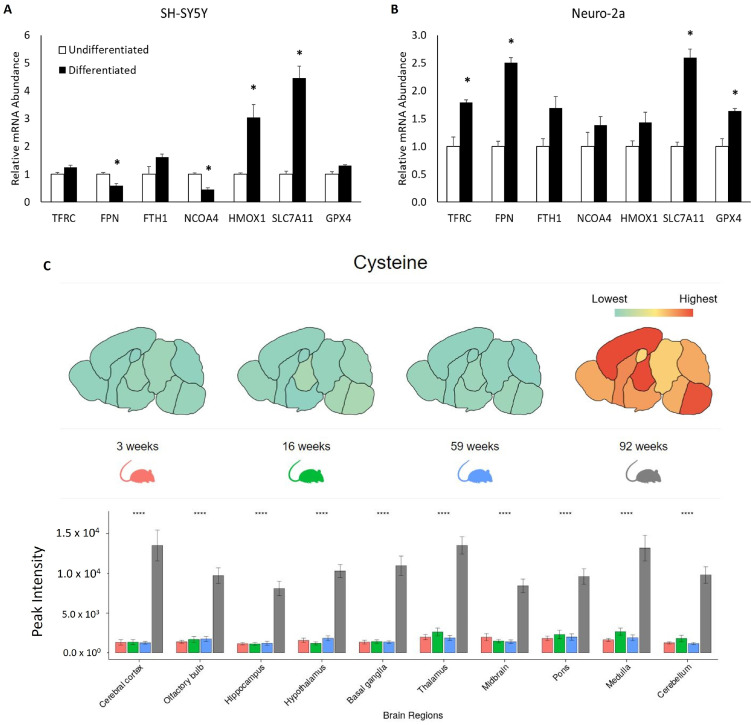
Neuronal differentiation and brain aging are associated with increased cysteine accumulation. Relative iron- and ferroptosis-related mRNA expression undifferentiated and differentiated (**A**) SH-SY5Y and (**B**) Neuro-2a cells. * Indicates differential expression from undifferentiated cells as analyzed by Student’s *t*-test (*p* < 0.05). (**C**) Cysteine levels in 10 different brain regions from aged mice (https://mouse.atlas.metabolomics.us/; accessed 8 August 2024). One-way ANOVA was used for significance analysis, * *p* < 0.05; **** *p* ≤ 0.0001.

**Figure 5 cells-13-01541-f005:**
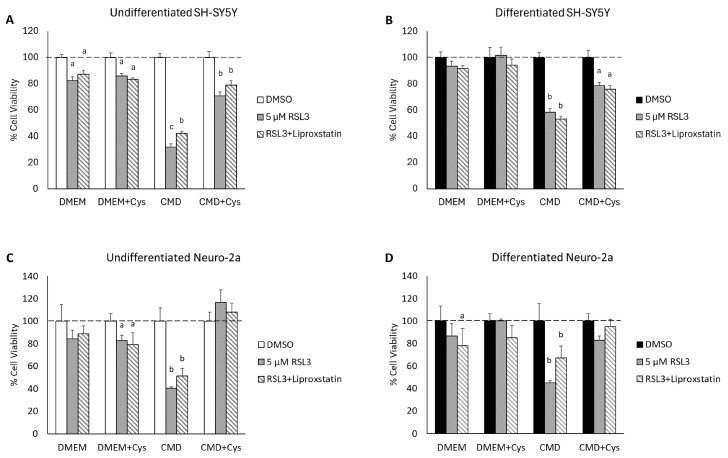
The influence of cysteine deprivation on the susceptibility to RSL3-mediated ferroptotic cell death in undifferentiated and differentiated neuroblastoma cell lines. Relative cell viability of (**A**) undifferentiated and (**B**) differentiated SH-SY5Y as well as (**C**) undifferentiated and (**D**) differentiated Neuro-2a cells in response to treatment with RSL3 when grown in control DMEM or cysteine and methionine-depleted DMEM (CMD). To indicate that the increased sensitivity to ferroptosis induction was a direct result of cysteine depletion, cells were rescued from cell death by supplementation with 200 µM cystine (+Cys). Superscripts indicate differential expression as analyzed by one-way ANOVA using Tukey’s post hoc test for differences in treatment response within a given media group (i.e., DMEM, DMEM + Cys, etc.) (*p* < 0.05).

**Table 1 cells-13-01541-t001:** Human primer sequences used for qPCR.

Gene Symbol	Accession Number	Forward Primer	Reverse Primer	Source
*FTH1*	NM_002032.2	5′aacatgctgagaaactgatg	5′gcacactccattgcattcagc	[18]
*GPX4*	NM_001039847	5′acaagaacggctgcgtggtgaa	5′gccacacacttgtggagctaga	[19]
*HMOX1*	NM_002133.1	5′cgggccagcaacaaagtg	5′agtgtaaggacccatcggagaa	Primer Express
*NCOA4*	NM_001145260.1	5′cagcagctctactcgttattgg	5′tctccaggcacacagagact	[20]
*PPIB*	NM_000942.5	5′tgccatcgccaaggagtag	5′tgcacagacggtcactcaaa	Primer Express
*SLC7A11*	NM_014331.4	5′atgcagtggcagtgaccttt	5′ggcaacaaagatcggaactg	[21]
*SLC40A1 (FPN)*	NM_014585.5	5′tgaccagggcgggaga	5′agaggtcaggtagtcggcca	[22]
*TFRC*	NM_001128148.1	5′agttgaacaaagtggcacgagcag	5′agcagttggctgttgtacctctca	[23]

**Table 2 cells-13-01541-t002:** Mouse primer sequences used for qPCR.

Gene Symbol	Accession Number	Forward Primer	Reverse Primer	Source
*Fth1*	NM_010239.2	5′tgatgaagctgcagaaccag	5′gtgcacactccattgcattc	[24]
*Gpx4*	NR_110342.1	5′acccactgtggaaatggatga	5′ctctatcacctggggctcctc	[25]
*Hmox1*	NM_010442	5′tcaggtgtccagagaaggcttt	5′tcttccagggccgtgtagat	Primer Express
*Ncoa4*	NM_019744	5′cgccagaccatcaccacat	5′ctcgcgtgagccatcagat	Primer Express
*Rpl19*	NM_000981	5′gacggaagggcaggcatatg	5′tgtggatgtgctccatgagg	Primer Express
*Slc7a11*	NM_011990.2	5′atctcccccaagggcatact	5′gcataggacagggctccaaa	[26]
*Slc40a1* *(Fpn)*	NM_016917.2	5′gctgctagaatcggtctttggt	5′cagcaactgtgtcaccgtcaa	[27]
*Tfrc*	NM_011638	5′ttggacatgctcatctaggaactg	5′ctgagatggcggaaactgagt	Primer Express

## Data Availability

The original contributions presented in the study are included in the article/Appendix A, further inquiries can be directed to the corresponding author.

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
