# Peer review of "Considerations for Using Neuroblastoma Cell Lines to Examine the Roles of Iron and Ferroptosis in Neurodegeneration"

_cells, 2024, doi:10.3390/cells13181541_

Round 1
Reviewer 1 Report
Comments and Suggestions for Authors
The present study explores neuronal iron metabolism and susceptibility to ferroptosis examining new evidence on the system xc-/glutathione peroxidase 4 (GPX4)/glutathione and emphasizing its upregulation as a potential mechanism by which aging neurons might protect. The authors used two neuroblastoma cell lines such as human SH-SY5Y and mouse Neuro-2a cells which were terminally differentiated into mature neurons followed by typical cellular and biochemical assays measuring specific mRNA and protein levels. I think the manuscript is well written, I have a few minor comments that I would like to be addressed.
The abstract is too extensive and the main point is lost. It should be reorganized and limited by emphasizing the main points
The authors should add in the legends which type of statistical method was perform. For example, the labelling is present in Figure 4 (One-way ANOVA), not in the other figures (Student’s test or other;)
In order to observed that the increased sensitivity to ferroptosis induction was a direct result of cysteine depletion, cells were rescued from cell death by supplementation with 200 μM cystine. How this concentration was selected? Was there any previous screening?
The mechanism of neuroblastoma cell lines differentiation upon all-trans retinoic acid could be further stress. Is 20 μΜ exposure sufficient for both cell lines?
The conclusions are to be made stronger; they are general in the current version. Some numerical results might be presented and the future directions are to be discussed in details.
Author Response
Reviewer 1 Comments:
The present study explores neuronal iron metabolism and susceptibility to ferroptosis examining new evidence on the system xc-/glutathione peroxidase 4 (GPX4)/glutathione and emphasizing its upregulation as a potential mechanism by which aging neurons might protect. The authors used two neuroblastoma cell lines such as human SH-SY5Y and mouse Neuro-2a cells which were terminally differentiated into mature neurons followed by typical cellular and biochemical assays measuring specific mRNA and protein levels. I think the manuscript is well written, I have a few minor comments that I would like to be addressed.
The abstract is too extensive and the main point is lost. It should be reorganized and limited by emphasizing the main points
Response: Thank you for your advice on making our abstract more reader friendly and concise. We have removed any unnecessary language that detracts from the key points made in the article.
The authors should add in the legends which type of statistical method was perform. For example, the labelling is present in Figure 4 (One-way ANOVA), not in the other figures (Student’s test or other;)
Response: We apologize for the original oversight. All figure legends have been updated to indicate the statistical tests performed for each experiment.
In order to observed that the increased sensitivity to ferroptosis induction was a direct result of cysteine depletion, cells were rescued from cell death by supplementation with 200 μM cystine. How this concentration was selected? Was there any previous screening?
Response: We appreciate the reviewer’s comment to increase the transparency in our decision making behind our rescue experiments. No prior screening was done. The 200 µM dose was selected because that is the concentration of cystine supplemented in standard commercial DMEM that the cells were grown in under control conditions, which allowed us to make comparisons to previous results. This explanation has been added to lines 341-343.
The mechanism of neuroblastoma cell lines differentiation upon all-trans retinoic acid could be further stress. Is 20 μΜ exposure sufficient for both cell lines?
Response: We believe that an in-depth discussion of the mechanisms involved in ATRA-mediated cell line differentiation is beyond the scope of this article. However, an additional citation on the specific mechanisms and brief mention of the capacity of ATRA to inhibit cell growth while simultaneously promoting differentiation are now provided in lines 88-90.
The conclusions are to be made stronger; they are general in the current version. Some numerical results might be presented and the future directions are to be discussed in details.
Response: We appreciate the reviewer’s suggestion to strengthen our conclusions section, but this work was submitted for consideration in the special issue of “Cell Method,” and as such the emphasis of the work was on methodological approaches and considerations for investigating neuronal ferroptosis in an in vitro model. Thus, by the nature of the experimental design, we believe we are restricted in the strength of the conclusions we can draw from our current work. We do believe, however, that readers will be able to use this work as a guide for future investigations that will advance the knowledge of this field. Additional suggestions for future studies are made in lines 411-414.
Reviewer 2 Report
Comments and Suggestions for Authors
There were some revisions in the present manuscript.
Introduction; background about ferroptosis might be insufficient. Line 70, ROS should be spelled out and the mechanism of ferroptosis would be better to add more details.
Line 128-129, the font was mistaken.
Line 182-183 "the addition of supplemental iron only statistically significantly increased ferritin protein expression in undifferentiated cells (Figures 1E and F)"; it was fact in SH-SY5Y cells, however, in Neuro2a cells the significant difference was not present in Figure 1F (DMSO vs Hemin in Undifferentiated). Which was correct between the text and Figure 1F?
Figures 1, 2, and 3, the images of morphology were not clear. The morphological changes (like the authors wrote in line 257-259) could not be assessed. Enlarged photos with high magnification also should be added.
Line 238-241, the data of erastin was not shown in the present manuscript. "Data not shown" should be added.
Line 252, correct "undifferenciated" to "Differentiated".
Line 259, correct "Figures 3D and F" to "Figures 3D and H".
Line 283, correct "Figure 1C-E" to "Figure 1C-F".
Subtitle of 3.5. 'Cysteine depletion sensitizes differentiated neurons to ferroptosis'; it was fact that RSL3 sensitivity was increased in CMD media. However, because liproxstatin-1 failed to rescue RSL3-mediated cell death, was not the cell death 'ferroptosis'? Was 'RSL3-induced cell death' better? Again, the explanation of 'ferroptosis' in Introduction would be written in more detail.
This is just my interest but is the incidence of iron accumulation in brain and AD/ADRD different in the conditions with iron deficiency in whole body such as anemia?
Author Response
There were some revisions in the present manuscript.
Introduction; background about ferroptosis might be insufficient. Line 70, ROS should be spelled out and the mechanism of ferroptosis would be better to add more details.
Response: We thank the reviewer for their many helpful suggestions throughout the manuscript. We have added more relevant information related to ferroptosis in the introduction and spelled out ROS in lines 41-44.
Line 182-183 "the addition of supplemental iron only statistically significantly increased ferritin protein expression in undifferentiated cells (Figures 1E and F)"; it was fact in SH-SY5Y cells, however, in Neuro2a cells the significant difference was not present in Figure 1F (DMSO vs Hemin in Undifferentiated). Which was correct between the text and Figure 1F?
Response: Thank you for pointing out this point of miscommunication. The figure was correct. The text has been modified accordingly to accurately describe the results as depicted in the figures (Lines 188-190).
Figures 1, 2, and 3, the images of morphology were not clear. The morphological changes (like the authors wrote in line 257-259) could not be assessed. Enlarged photos with high magnification also should be added.
Response: We agree with the reviewer that the images in the uploaded Word document in the original submission were not clear. High resolution images for each figure have been uploaded. As this manuscript will only be published online, the availability of these high-resolution images will be helpful in supporting the conclusions drawn in the text.
Line 128-129, the font was mistaken.
Line 238-241, the data of erastin was not shown in the present manuscript. "Data not shown" should be added.
Line 252, correct "undifferenciated" to "Differentiated".
Line 259, correct "Figures 3D and F" to "Figures 3D and H".
Line 283, correct "Figure 1C-E" to "Figure 1C-F".
Response: We appreciate the reviewer’s keen eye and helpfulness in optimizing the clarity of our work. Appropriate corrections to all 5 points above have been made.
Subtitle of 3.5. 'Cysteine depletion sensitizes differentiated neurons to ferroptosis'; it was fact that RSL3 sensitivity was increased in CMD media. However, because liproxstatin-1 failed to rescue RSL3-mediated cell death, was not the cell death 'ferroptosis'? Was 'RSL3-induced cell death' better? Again, the explanation of 'ferroptosis' in Introduction would be written in more detail.
Response: We agree with the reviewer and apologize for this initial overstatement. The subtitle has been modified to more accurately reflect the description of the results and the discussion. As stated above, we have attempted to better explain ferroptosis in the introduction to add clarity to our conclusions.
This is just my interest but is the incidence of iron accumulation in brain and AD/ADRD different in the conditions with iron deficiency in whole body such as anemia?
Response: This is a good question, particularly as higher levels of iron in the brain have long been found in individuals with diagnosed AD. But whole-body iron status does not appear to be a strong indicator of AD/ADRD risk as indicated by the evidence that women, who tend to have much higher rates of anemia are not at a reduced risk for AD/ADRD. However, one study has indicated that males with hereditary iron overload are at increased risk AD/ADRD (https://www.ncbi.nlm.nih.gov/pmc/articles/PMC7990419/). In individuals without hemochromatosis however, the accumulation of iron in the brain appears to be more of a pathologic consequence of the other metabolic perturbances that underly AD/ADRD pathology.
Round 2
Reviewer 2 Report
Comments and Suggestions for Authors
The authors replied to the reviewer's comments and improved the manuscript.